# Proteomic Comparison of Ivermectin Sensitive and Resistant *Staphylococcus aureus* Clinical Isolates Reveals Key Efflux Pumps as Possible Resistance Determinants

**DOI:** 10.3390/antibiotics11060759

**Published:** 2022-06-02

**Authors:** Shoaib Ashraf, Débora Parrine, Muhammad Bilal, Umer Chaudhry, Mark Lefsrud, Xin Zhao

**Affiliations:** 1Department of Animal Sciences, McGill University, Sainte-Anne-de-Bellevue, Quebec, QC H9X 3V9, Canada; muhammad.bilal@mail.mcgill.ca; 2Wellman Center for Photomedicine, Massachusetts General Hospital, Harvard Medical School, Boston, MA 02114, USA; 3Department of Pathobiology, Riphah University, Lahore 54500, Pakistan; 4Department of Bioresource Engineering, McGill University, Sainte-Anne-de-Bellevue, Montreal, QC H9X 3V9, Canada; debora.parrine@imbim.uu.se (D.P.); mark.lefsrud@mcgill.ca (M.L.); 5Biomedicinska Centrum, Institutionen för Medicinsk Biokemi och Mikrobiologi, Uppsala Universitet, 751 23 Uppsala, Sweden; 6School of Veterinary Medicine, University of Surrey, Guildford GU2 7AL, UK; u.chaudhry@surrey.ac.uk

**Keywords:** ivermectin, *Staphylococcus aureus*, proteomics, genomics, efflux pumps, virulence factor

## Abstract

Ivermectin (IVM) is a versatile drug used against many microorganisms. *Staphylococcus aureus* is one of the most devastating microorganisms. IVM sensitive and resistant *S*. *aureus* strains were recently reported. However, the underlying molecular mechanisms of resistance are unknown. Clinical isolates of *S. aureus* were used for determination of the sensitivities against IVM by growth curve analysis and time-kill kinetics. Then, proteomic, and biochemical approaches were applied to investigate the possible mechanisms of resistance. Proteomic results showed a total of 1849 proteins in the dataset for both strains, 425 unique proteins in strain O9 (IVM sensitive), and 354 unique proteins in strain O20 (IVM resistant). Eight proteins with transport functions were differentially expressed in the IVM resistant strain. Among them, three efflux pumps (*mepA*, *emrB*, and *swrC*) were confirmed by qPCR. The IVM resistant *S*. *aureus* may overexpress these proteins as a key resistance determinant. Further experiments are required to confirm the exact mechanistic relationship. Nevertheless, the possibility of blocking these transporters to reverse or delay the onset of resistance and reduce selection pressure is potentially appealing.

## 1. Introduction

Ivermectin (IVM) is classically an anti-parasitic macrocyclic lactone (ML) that acts on glutamate-gated chloride channels to induce hyperpolarization/paralysis of the cells, and thus results in death of the cell/parasite [1]. IVM is known as a miracle drug since it has shown activities against a wide range of targets [2]. For example, IVM inhibits sporogony and shows an activity against *Plasmodium falciparum* [3]. More promisingly, IVM has exhibited in vitro antiviral activities against a wide range of RNA and DNA viruses, including avian influenza A virus, dengue virus, West Nile virus, human immunodeficiency virus-1 (HIV), Venezuelan equine encephalitis virus, and pseudorabies virus [4,5]. In addition, IVM has shown anticancer potential in several studies by either inhibiting the tubulin proteins from disassembling (disturbing the tubulin equilibrium) or acting as an antimitotic agent or by chloride dependent membrane hyperpolarization [6,7,8,9]. Further, IVM has shown an activity against a number of bacteria, including the resistant *Mycobacterium tuberculosis* and *Mycobacterium ulcerans* [10,11]. Clinically, IVM has been used against different helminth and other parasitic infections both in animal and human medicine throughout the globe.

*Staphylococci* are Gram-positive cocci having more than 30 species and among them *Staphylococcus aureus* (*S*. *aureus*) is the most devastating [4]. The significance of *S*. *aureus* can be envisaged by the fact that more than 30% of the population across the globe are silent carriers [12]. Although most of these individuals are asymptomatic, *S*. *aureus can* initiate many disease pathologies ranging from skin infections to severe life-threatening diseases in a wide variety of host species and *S*. *aureus is* the leading cause of bacteremia in humans [13]. To this end, World Health Organization (WHO) has classified *S. aureus* resistant to methicillin and vancomycin as one of the microorganisms that immediately require new antibiotics for successful therapy. *S. aureus* infections are also of significant importance in the animal health sector as it is one of the most common microorganisms that causes mastitis in the dairy industry [14]. Infections by *S*. *aureus* are prevalent all over the world and treatment of this notorious bug has been difficult due to emergence of antibiotic resistance. β-lactams were extensively used to treat *S*. *aureus* infections as the first line of defense in the past. However, they have slowly lost efficacy. Due to widespread resistance against them, a semi-synthetic penicillinase resistant β-lactam, i.e., methicillin, was introduced as an alternative to overcome the cleavage of these natural β-lactams by β-lactamases. Unfortunately, resistance developed against methicillin soon after its introduction [15]. Consequently, methicillin resistant *Staphylococcus aureus* (MRSA) are prevalent throughout the globe [16] and they are often resistant to most of the known antibiotics. To this end, different efflux pump inhibitors have been explored as a strategy to overcome antibiotic resistance in *S. aureus* species [17]. Recently, our group has shown that IVM has an activity against clinical isolates of *S*. *aureus* [4]. However, approximately 10% of isolates were resistant to IVM. We thus sought to decipher the possible mechanism behind this unprecedented observation. Hence, in the present study we aimed to dissect the molecular mechanisms of resistance of IVM against *S*. *aureus* (with a focus on efflux pumps/transporters) and the results might open new avenues to prolong the efficacy of IVM.

## 2. Materials and Methods 

### 2.1. Bacterial Isolates and Resistance Classification

Two *S. aureus* clinical isolates (O9 and O20) were used in the present study. The isolates were isolated and identified as *S*. *aureus* from mastitis cases from Lahore, Pakistan. Following initial isolation and identification the isolates were shipped to McGill University, Canada for further analyses. The resistance pattern for antibiotics was performed by the disc diffusion method, except for cefoxitin. Both strains were sensitive to methicillin, and cefoxitin (MIC 1 μg/mL) classifying them as methicillin sensitive *Staphylococcus aureus* (MSSA). In addition, both strains were resistant to tetracyclines (zone inhibition diameter < 14 mm), and erythromycin (zone inhibition diameter < 18 mm). O9 was sensitive to norfloxacin (zone inhibition diameter > 17 mm) and ciprofloxacin (zone inhibition diameter > 21 mm), while O20 was resistant to these two fluoroquinolones (norfloxacin zone inhibition diameter < 12 mm; ciprofloxacin zone inhibition diameter < 15 mm). The bacteria were grown in Cation-adjusted Mueller Hinton (MH) broth and tryptic soya broth (TSB) (Sigma Aldrich, Oakville, ON, Canada) at 37 °C. There are no standard guidelines developed by the Clinical and Laboratory Standard Institute (CLSI) for the use of IVM against bacteria. Thus, for this study we arbitrarily defined the sensitive strain as the one which could be killed by IVM at the concentration of ≥12.5 μg/mL, while the resistant strain was not killed by IVM up to 100 μg/mL concentration. 

### 2.2. Effects of IVM on Bacterial Growth

To determine the effects of IVM on bacteria using the growth curves [18], the 0.5 McFarland culture of both isolates was prepared and further diluted to obtain a final concentration of 5 × 10^5^ CFU/mL. IVM dissolved in dimethylsulfoxide (DMSO) (solvent control) was then serially titrated at concentrations of 100, 50, 25, 12.5, 6.25, 3.125, and 1.56 μg/mL. The optical density (O.D) values were taken at 2 h intervals for a total of 14 h at 37 °C. The O.D value was measured at 600 nm in a spectrophotometer with 1 mL cuvettes. All the bacteria were grown in a shaking incubator at 200 rpm and 37 °C. At least three biological and technical replicates were performed for each experiment throughout the study.

### 2.3. Time Kill-Kinetics

For further confirmation of the above experiments, 0.5 McFarland bacterial suspensions were prepared for both isolates from freshly prepared overnight cultures as described above by time kill-kinetics [19]. Then, both *S*. *aureus* isolates were further diluted in MH broth to attain a final concentration of 5 × 10^5^ CFU/mL. Following this, IVM was added at concentrations of 12 × (MIC) (150 μg/mL), 8 × (MIC) (100 μg/mL), 4 × (MIC) (50 μg/mL), 2 × (MIC) (25 μg/mL), 1 × (MIC) (12.5 μg/mL), ½ × (MIC) (6.25 μg/mL), ¼ × (MIC) (3.125 μg/mL), and 0 × (MIC) (1.562 μg/mL), for O9 IVM sensitive isolate, whereas up to 100 μg/mL for O20 (IVM resistant) strain. The tubes were then incubated at 37 °C and 100 μL samples were taken at 0, 2, 4, 8, and 12 h. The aliquots were then plated onto MH agar plates to assess viable bacteria by CFU counting. After 24 h of incubation at 37 °C, plates were examined for growth. 

### 2.4. Protein Extraction and Digestion

Whole proteome analysis was performed, as described below. Bacterial cell lysis, protein extraction digestion, reduction, and alkylation were carried out, as previously described [20]. Briefly, bacterial cells were harvested by centrifugation at 4 °C for 15 min at 21,000× *g*. The resulting pellets (100 mg for each sample) were then solubilized in a buffer (100 mM Tris/4% SDS, pH 8). To improve cell lysis, the samples were sonicated with pulses of 10 s and 20 s on/off for a total of 5 min in cold water to avoid heating. Next, the samples were centrifuged at 4 °C for 10 min at 21,000× *g* to remove insoluble material. The supernatant containing proteins was reduced with 20 mM TCEP (Tris(2-carboxyethyl) phosphine) (Sigma-Aldrich, Oakville, ON, Canada). Non-protein contaminants were removed by 20% trichloroacetic acid/acetone precipitation. Proteins in the precipitate were resolubilized by adding 8 M urea in Tris-HCl pH 8.0 for 30 min at room temperature. To avoid trypsin inhibition in the following steps, the urea concentration was reduced to 2 M urea. Protein reduction was obtained by adding 20 mM TCEP to the sample for 30 min at room temperature. Protein alkylation was achieved by adding 20 mM iodoacetamide to the sample and incubation at room temperature in the dark for 30 min. Protein concentration was determined using a BCA protein assay kit (Thermo Fisher Scientific, Waltham, MA, USA). Proteins (1–2 mg) were digested with modified sequencing grade trypsin (Promega, Madison, WI, USA), for 12 h at 37 °C. Then, an acidic solution (200 mM NaCl, 0.1% formic acid) was added to stop trypsin activity. Trypsin and undigested proteins were removed by a 30 kDa MWCO Amicon ultra-centrifugal filter (Millipore Sigma, Middlesex County, MA, USA) [21,22,23,24]. The filtrate containing the digested proteins (<30 kDA) was used for further analysis and the peptides were desalted by a centrifugal column (Sep-Pak Plus C-18, Waters Limited, Mississauga, ON, Canada) before peptide quantification (Pierce Quantitative Colorimetric Peptide Assay, Thermo-Fisher Scientific, San Jose, CA, USA) and stored at −80 °C.

### 2.5. Mass Spectrometry Analysis (2D LC−MS/MS) 

A Multi-dimensional Protein Identification Technology (MudPIT) approach was adopted to obtain a label-free shotgun proteomic analysis [21]. A high-performance separation of the peptides was obtained by using a 2D-LC separation coupled online with the mass spectrometer (LTQ XL, Thermo Fisher Scientific, San Jose, CA, USA). The LC system solvents consisted of 0.1% formic acid in water and 0.1% formic acid in acetonitrile. To generate a peptide fractionation, 12 unique gradients of the LC solvents were applied to the column for 90 min each. The gradients consisted of a mixture of buffer A (H_2_O, 0.1% formic acid) and buffer B (CH_3_CN, 0.1% formic acid). Each gradient consisted in the following concentrations of buffer B: 0%, 5%, 7.5%, 10%, 12.5%, 15%, 17.5%, 20%, 25%, 35%, 50%, 100%, in combination with buffer A. Then, 60 µg of peptides were bomb-loaded through cell-pressure chamber into a biphasic column packed with ~5 cm of strong cation exchange (SCX) resin and ~5 cm of C18 reversed phase (RP) material (Luna 5 μm 100 A and Aqua 5 μm 100 A, respectively, Phenomenex, Torrance, CA, USA). Packed column was washed through the cell-pressure chamber for 60 min with H_2_O (MS-grade Optima, Thermo Fisher Scientific) with 0.1% formic acid to remove salts and impurities. Peptide spray was generated by a front column containing an integrated nanospray-emitter tip (100 μm i.d., 360 μm o.d., 15 μm i.d. tip, New Objective, Woburn, MA, USA) loaded with ~15 cm of C18 material. A 12-step gradient containing salt pulses was utilized to elute the peptides from the columns in a nanoESI-MS/MS approach. The data-dependent acquisition parameters used in Xcalibur (v.2.0.7 SP1) were: the 5 most intense MS/MS were submitted to collision-activated dissociation (35% energy) after two microscans for both full and MS/MS scans with 3 m/z isolation width and a dynamic exclusion repeat of 1 for 30 s, centroid data for all scans and two microscans averaged for each spectrum [22].

### 2.6. Database Searching

Thermo RAW files were used to extract the MS/MS spectra which were searched against a database containing target and reverse peptide sequences of *S. aureus* (UNIPROT, proteome UP000187748) and common contaminants (cRAP v. 2012.01.01). In a multiple search engines approach [23], SEQUEST HT [24] and, MSAmanda) [25] algorithms were used for protein identification through the software Proteome Discoverer v.2.1.1. (Thermo Fischer Scientific, Inc.). The search parameters included a maximum of 2 missed cleavages, parent ion and fragment tolerance of 2.5 Da and 0.5 Da, respectively. Methionine oxidation (+15.99 Da) was set as a variable modification, and carbamidomethylation of cysteines (+57.05 Da) was kept as a static modification in all search engines. Peptide length was set to ≥6 and a strict false discovery rate was controlled at ≤1% at peptide and PSM levels. Protein identifications and expression values expressed as emPAI [26]. The results were imported into Perseus [27] and the emPAI values were normalized across samples. To further enhance the result confidence, only proteins containing fold changes of ≥1.65 (differentially expressed) or unique proteins containing PSM ≥ 3 were considered in the analysis.

### 2.7. Bioinformatics

Functional annotations of the identified proteins were obtained via the UNIPROT Gene ontology tool (UNIPROT-GOA) [28], which includes annotations from: InterPro2GO, UniProt keywords2GO, Enzyme Commission2GO, and UniPathway2GO. Molecular function and biological processes GO term were investigated. Since UNIPROT-GOA offers automated annotation, the functions of candidate proteins were further verified, using the literature. Protein interaction analysis was performed by mapping UNIPROT accessions to gene names and searching them in the STRING database (version 11.5) versus the *S. aureus* protein interactions. Only interactions with a high confidence (minimum score of 0.70) were maintained in the network. The sources utilized for the generation of the networks were from neighborhood, experiments, gene fusion, databases, co-occurrence, and co-expression (textmining was excluded to increase accuracy). Disconnected nodes were not displayed in the networks.

### 2.8. Extraction of RNA

The *S*. *aureus* strains (O9 and O20) were obtained from glycerol stocks at −80 °C and grown in MH broth overnight at 35 °C. After overnight growth the bacteria were pelleted at 2000× *g* for 10 min. RNA was extracted through PureLink RNA Mini Kit (Ambion Life Technologies, Invitrogen, Mississauga, ON, Canada) and treated with DNase enzyme (PureLink, Invitrogen, Mississauga, ON, Canada). The RNA was quantified by Nanodrop (Denovix-DS-11-FX) and RNA integrity was determined through gel electrophoresis. The RNA samples with A260/A280 ratio greater than 1.8 and sharp bands of 23S and 16S were selected for downstream application and stored at −80 °C.

### 2.9. Reverse Transcription and Quantitative PCR

For cDNA syntheses a total of 100 ng of RNA was used for each sample. The RNA was reverse transcribed into cDNA using the High-Capacity cDNA Reverse Transcription kit (as per manual instruction with final volume of 20 μL/reaction (Applied Biosystems, Mississauga, ON, Canada). The cDNA samples were stored at −20 °C for further use. Quantitative PCR was performed using SsoAdvanced Universal SYBR Green Supermix (Biorad, Mississauga, ON, Canada) using the Biorad CFX Connect 96 well Real Time PCR System. The information about the primers for different bacterial transporters (target genes) are shown in Table 1 (Invitrogen). The final volume per reaction was kept 10 μL and each reaction was run in triplicate. The qPCR was set to 30 s at 95 °C for polymerase activation and 5 s at 95 °C for denaturation. The annealing temperature was set at 60 °C for 30 s and repeated the procedure for 40 cycles. The melt curves were set between 65 °C and 95 °C, with an 0.5 °C increment after every 3 s. Gel electrophoresis was used to check the specificity of PCR products. The gene, *gyraseA* of *S. aureus* was used as reference (housekeeping gene) and reaction values of target genes (*mepA, emrB,* and *swrC*) were normalized with the reference gene and results were shown as normalized expression. 

## 3. Results

### 3.1. Inhibition of S. aureus Growth by IVM as Evidenced by Growth Curves and Time-Kill Kinetics

The growth curve analysis demonstrated that IVM reduced the growth of O9, even at the low concentration of 1.56 μg/mL. On the other hand, IVM did not reduce the growth of O20 even at the high concentration of 100 μg/mL (Figure 1A,B). Similarly, the results of time-kill kinetics were the same as those of the growth curves. IVM showed reduction in CFU/mL as compared to the control (DMSO) for O9 but not for O20. The bacterial growth was completely inhibited by IVM at concentrations of ≥12.5 μg/mL for O9 whereas no inhibition was observed for the O20 strain in the presence of IVM up to 100 μg/mL (Figure 1C,D). Notably, as stated above since there are no established CLSI protocols for prediction of IVM resistance against *S. aureus*, hence for this study we have defined the sensitive and tolerant IVM strains as the ones which could be killed/survive in the presence of IVM at the concentrations of ≥12.5 or up to 100 μg/mL, respectively.

### 3.2. Proteomic Analysis

To observe if it is possible to distinguish why strains O9 and O20 were different towards their sensitivity to IVM, a two-shot proteomic analysis of *S. aureus* strains O20 and O9 was carried out. The protein contents of the bacterial isolates were analyzed by a LC-MS/MS MudPIT, which allows for a multidimensional analysis of the peptides (Figure 2).

To increase the chances for identification and quantitation of peptides, in the LC-MS/MS analysis, a total of 12 unique gradients of the LC solvents were applied. This approach decreases the effect of co-isolation of peptides, resulting in a larger number of proteins identified. A total of 1846 proteins were identified in the dataset of strains O9 and O20, representing about 64% of the proteins from the reference proteome in UNIPROT database (total of 2889 protein entries in 2022) for *S. aureus* strain NCTC/PS47 (proteome ID: UP000008816). Specifically, 425 were only found in strain O9 and, 354 in strain O20 (Figure 3). After normalization of the label-free quantitative values by the emPAI method, the fold change calculation of only the characterized proteins (strain O9/strain O20) was performed. 

DNA binding (7.0% vs. 5.37%), RNA binding (5.76% vs. 6.90%), metal ion binding (9.42% vs. 10.73%), nucleotide binding (16.93% vs. 19.61%), catalytic activity (46.11% vs. 46.11%), were the main functions in most proteins of O20 vs. O9 strains, respectively. Interestingly, the function of transporter activity was found in about 5.23% of proteins from strain O20, while very few proteins with the transporter activity were detected in strain O9 (Figure 4). A total of 458 proteins were differentially expressed in O20 versus the O9 strain (fold change ≥1.65). Upon a functional enrichment analysis, 93 of these proteins have roles in cellular processes, 91 involved in metabolic processes, 14 in biological regulation, 9 involved in response to stimulus, 9 involved in localization, 8 are efflux pumps for transportation, 2 in cytolysis, 1 in cell adhesion, and 1 in cytokinesis (Figure 4, and Appendix A (Protein Table)).

Next, we performed a protein interaction network analysis. We matched the list of differentially expressed proteins to the known interactions of *S. aureus* within the STRING database. The connecting proteins in the networks are known interactors either as gene neighborhood, gene fusion, co-occurrence, co-expression, or homology (Figure 5A,B). The network of the upregulated proteins (Figure 5A) presented five main clusters, with the densest gene cluster (cluster 1) containing proteins related to cytoplasmic translation. Protein functions from clusters 2–5 were de novo IMP biosynthetic process, ATP synthesis, ligase activity, unfolded protein binding, respectively. The network of the downregulated proteins (Figure 5B) presented only one significative cluster containing proteins (cluster 1) with the nucleic acid binding function. The remaining proteins are involved in organonitrogen compound catabolic process (*hutU* and *hutI* genes), and biological regulation (*leuS* and *valS* genes).

As our focus was to identify whether any transporters were involved in the IVM resistance phenotype so proteins only present or upregulated in strain O20 with roles in transportation are shown in Table 2. Proteins MsbA2, EmrB4, NorB3, and BTN44_13755 were only presented in the proteomic analysis of strain O20, while proteins EcsA3, YtrB1, MepA, and SwrC were more abundant (fold change >1.65) in strain O20. A list of all proteins quantified and normalized is presented in the Appendix A (Protein Table). A hierarchical clustering analysis of the protein abundance is reported as a heatmap in Figure 6. For a detailed view of Figure 6, an interactive version can be found in the file “Heatmap_SA.html”, in the Appendix A section.

### 3.3. Validation of the Overexpressed Bacterial Proteins

The mRNA quantification analysis by qPCR demonstrated that among the three target genes analyzed, *swrC*, *mepA,* and *emrB* were overexpressed in the O20 compared to the O9 strain (Figure 7). 

## 4. Discussion

IVM is primarily used as an anthelmintic in human and veterinary medicine. Previously, we reported its efficacy against *S*. *aureus*. Interestingly, among the many isolates investigated one isolate was resistant to the drug [4]. The growth curve analyses were in concert with the time-kill kinetics data as the O.D values did not increase beyond 12.5 μg/mL in O9 whereas the growth never stopped even when the concentration of IVM was increased up to 100 μg/mL for the resistant strain. The time-kill kinetics curves also showed inhibition of bacterial growth beyond 1× (MIC) in the sensitive strain whereas IVM was not able to inhibit bacterial growth in the resistant strain, even at the concentration of 100 μg/mL (Figure 1). 

Proteins that were only found in one of the samples did not present any fold change, as expected. Functional analysis revealed that most of the upregulated proteins had catalytic activity, followed by nucleotide ion binding, metal ion binding, DNA and RNA binding proteins. There were at least 354 proteins that were differentially expressed when compared between the two isolates (Figure 3). The group of 354 proteins that were upregulated have roles in cellular, metabolic, regulation, stimulus, localization, pumps, cytolysis, adhesion, and cytokinesis functions with the decreasing order in terms of numbers of proteins involved, respectively (Figure 4). Notably, the proteins MecA, BlaI, BlaZ, and UvrA which are involved in drug response were also differentially expressed. MecA targets unfolded and aggregated proteins to be recognized by proteolytic complexes. BlaZ, and BlaI are involved in the beta-lactam catabolic process, the latter is a transcriptional repressor, constitutively blocking the beta-lactamase gene. UvrA is part of the UvrABC complex that recognizes and processes the DNA lesions.

Efflux pumps are one of the key resistance determinants involved in IVM resistance against different parasites, so they were the focus of this study (Figure 7). To this end, the expression of *swrC* increased in the resistant *S. aureus* O20 strain in this study. SwrC is encoded by the gene *yerP* and is involved in the efflux of many substrates, including surfactin, acriflavine, and ethidium bromide to name a few [29,30]. The *swrC* is also classified as a pathogenicity related gene in *S. aureus* [31,32] while in other bacteria, such as *Bacillus subtilis*, it is reported to be linked with motility of the bacteria [33]. Its higher expression in resistant strain of *S. aureus* suggests increase in tolerance of bacteria to different noxious agents. The *emrB* gene was also significantly higher in the O20 strain. This gene was found to be active in expulsion of many compounds, such as oxidative salicylic acid, 2,4-dinitrophenol (DNP), and carbonyl cyanide m-chlorophenylhydrazone (CCCP), out of the cell membrane [34]. Zhang et al. [35] reported *emrB* involvement in resistance of *S. aureus* to ampicillin antibiotic and found that *emrB* was upregulated 2.34-folds after treatment with ampicillin while after knockout the ampicillin resistance decreased by 0.60-folds. Similarly, *mepA* is another transporter that belongs to the multidrug and toxic compound extrusion (MATE) family [36] and the protein expression was significantly higher in the O20 strain. It is previously observed [37] that overexpression of *mepA* in *S. aureus* led to multi-drug resistance against several monovalent and divalent biocides and the fluoroquinolone antimicrobial agents norfloxacin and ciprofloxacin. These observations reflect that *swrC*, *emrB,* and *mepA* may have played a key role in resistance phenotype of *S. aureus* against IVM and its higher virulence. However, involvement of other proteins than efflux pumps as the resistance determinants needs to be further investigated. Recently, an analogue of IVM has also shown potent efficacy against biofilms produced by MRSA strains [38]. This study reiterates the importance for the use of IVM as an anti-staphylococcal agent. 

To this end, we investigated the differences between two clinical isolates of *S*. *aureus* and attribute these disparities towards the overexpression of efflux pumps (transporters). Interestingly, such resistance mechanisms have also been reported for IVM against different type of anthelmintics. For example, it is known that IVM is a good substrate of efflux pumps, such as ABC transporters and P-glycoproteins, in different parasites [39,40]. Thus, it might be possible to use different blocking agents to occupy the drug binding pockets of these efflux pumps/transporters/proteins to reverse IVM resistance. IVM is used at a dose rate of 150–200 μg/kg body weight in humans and animals and at this dose the Cmax achieved is 50 ng/mL [41]. Although the concentration of IVM required to kill the sensitive strain in this study (12.5 μg/mL) was higher than its therapeutic use, the reported lethal dose 50 (LD_50_) of IVM (50 mg/kg) suggests that it has a wide therapeutic index [42]. The data hint that the transporters may play a critical role and using agents that block these transporters might help in reversing IVM resistance to not only *S*. *aureus* but other microorganisms as well. However, this clearly requires further investigation and was beyond the scope of this study. Moreover, the drug safety data show that IVM has a wide therapeutic window and further investigation of reversal of IVM resistance using efflux pump inhibitors might be an appealing option [42]. The current study despite being comprehensive has some limitations as we did not include any standard strains of *S. aureus*. Furthermore, the concentrations of IVM used in the current study may not be achievable under field conditions. Nevertheless, the findings shed light on an interesting possible mode of resistance to IVM. To this end follow-up work is needed with more isolates where we look for a deeper analysis to monitor the whole genome for mutations or in-trans complementation of the transporters involved in IVM resistance against *S. aureus* standard and clinical isolates. 

## 5. Conclusions

The present study investigated the proteomic differences between an IVM-sensitive *S*. *aureus* and an IVM-resistant *S. aureus* isolate. Upon proteomic and gene expression analyses, we found overexpression of important efflux pumps (transporters) in the IVM resistant isolate. The study reveals the possible potential mechanism of drug resistance of IVM, which can be further exploited to reverse and reduce selection pressure. 

## Figures and Tables

**Figure 1 antibiotics-11-00759-f001:**
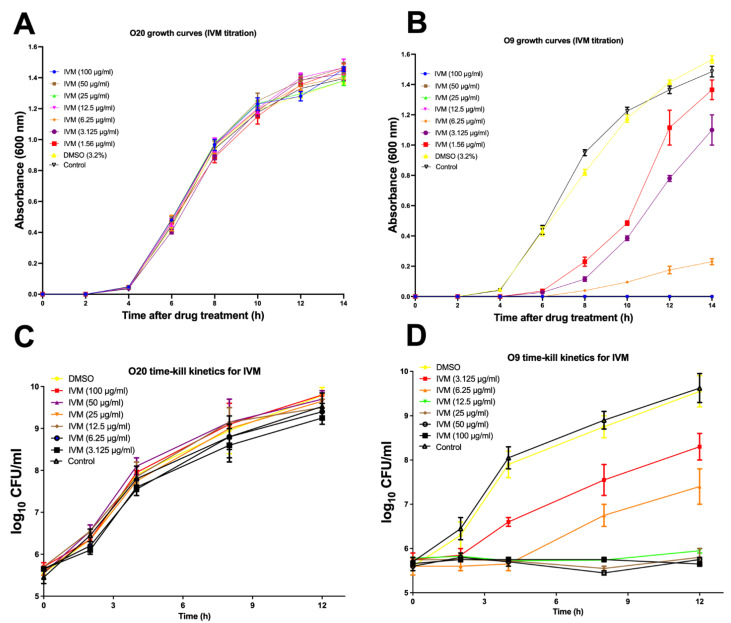
Growth curves and time-kill kinetics for O20 (**A**,**C**) and O9 (**B**,**D**) strains, respectively, in the presence of ivermectin. The control did not contain ivermectin or DMSO. The error bars are standard error mean for three independent experiments, each having three technical replicates, *n* = 9.

**Figure 2 antibiotics-11-00759-f002:**
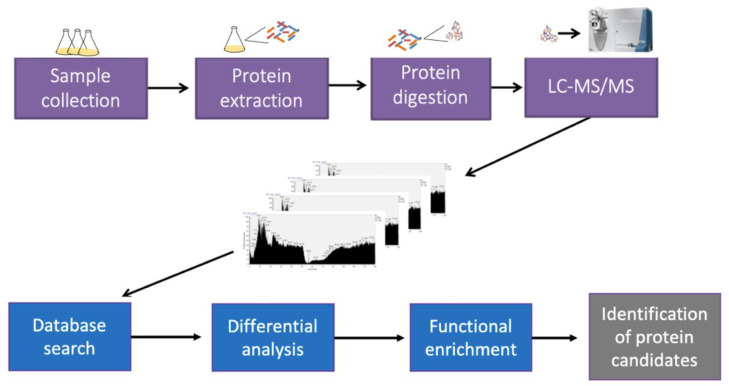
Experimental workflow of the proteomic analysis showing the steps for sample preparation, quantification by LC-MS/MS (MudPIT) and bioinformatics. The susceptible strain O9 was compared to the resistant strain O20 by label-free quantitation (emPAI).

**Figure 3 antibiotics-11-00759-f003:**
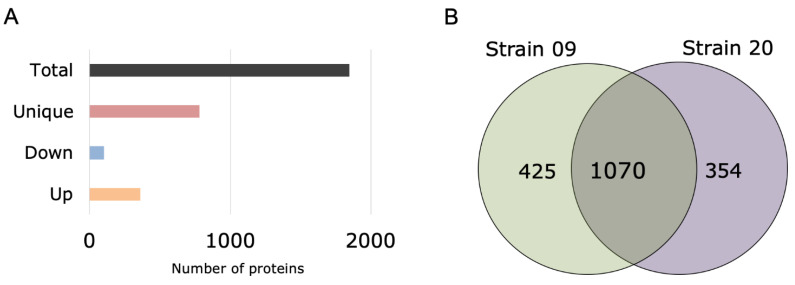
Distribution of the differently abundant and unique proteins found in strains O9 and O20. Number of proteins found in strain O20 that are unique, upregulated, downregulated, and in total (accounting for unchanged expression) in both strains O9 and O20 (**A**). The Venn diagram of proteins identified uniquely in strains O9 and O20, and in both (**B**).

**Figure 4 antibiotics-11-00759-f004:**
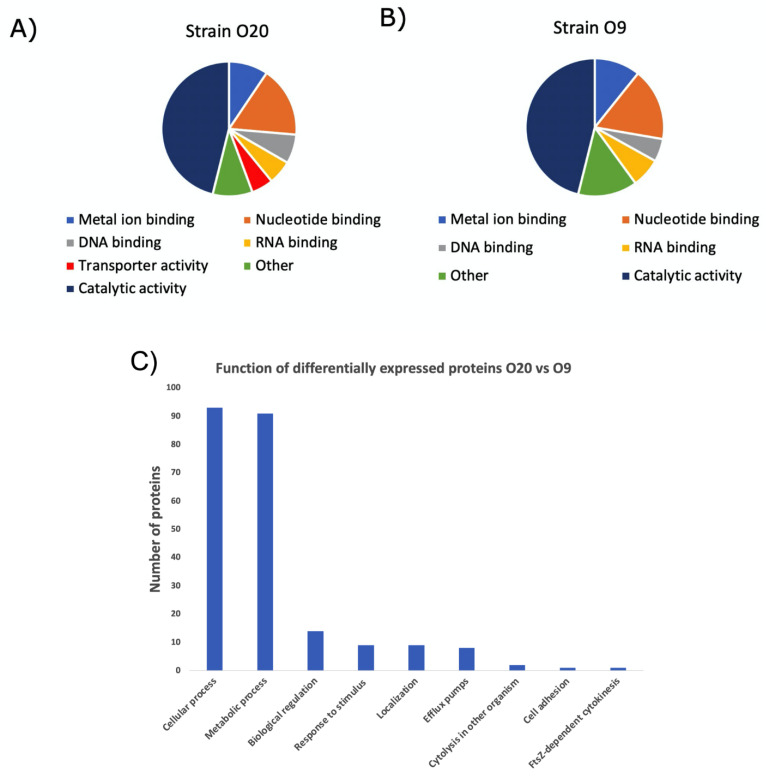
Graphs of the functional annotation of all proteins identified in strain O20 (**A**) and strain O9 (**B**). Functions of deferentially expressed proteins in O20 vs. O9 strain (**C**). GO terms assigned are associated with molecular function from UNIPROT-GOA. Metal and nucleotide-binding proteins are a subdivision of catalytic activity.

**Figure 5 antibiotics-11-00759-f005:**
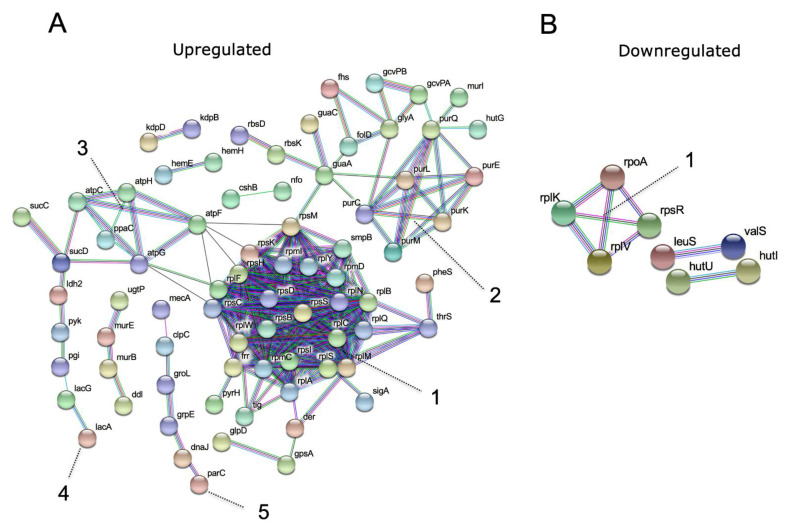
Protein interaction networks of differently expressed proteins in strain O20 versus O9. Protein accessions were mapped to gene names, their interactions are shown in the networks of upregulated proteins presented as five clusters, with cluster 1 containing proteins related to cytoplasmic translation. Cluster 2–5 have de novo IMP biosynthetic process, ATP synthesis, ligase activity, unfolded protein binding functions, respectively. (**A**) and downregulated genes, cluster 1 is the significant cluster containing proteins with the nucleic acid binding function (**B**). Only high confidence (minimum score 0.7) interactions are shown. Edges color scheme: gene neighborhood, green; gene fusions, red; co-occurrence, dark blue, co-expression, black; homology, light violet; data from curated databases, light blue, data experimentally determined, pink.

**Figure 6 antibiotics-11-00759-f006:**
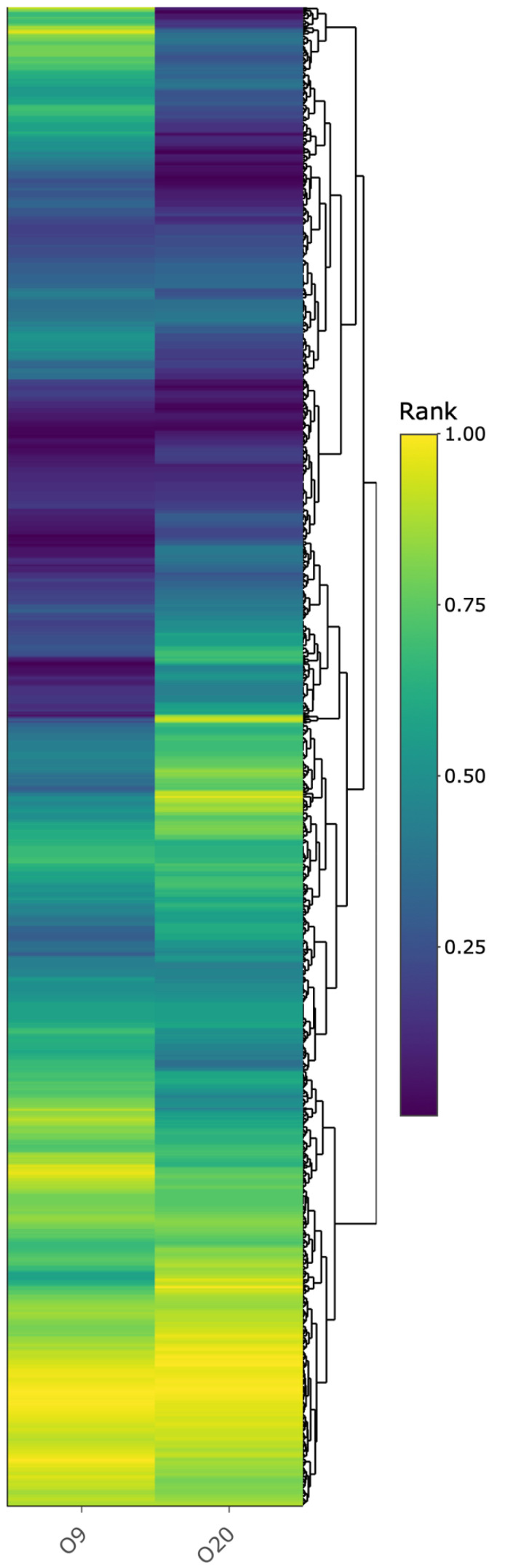
Heatmap depicting protein abundance after unsupervised hierarchical clustering of the data from *S. aureus* strains O9 and O20 containing the proteins normalized by rank from 0 to 1. An interactive version of this heatmap can be accessed in the Appendix A (Heatmap_SA.html).

**Figure 7 antibiotics-11-00759-f007:**
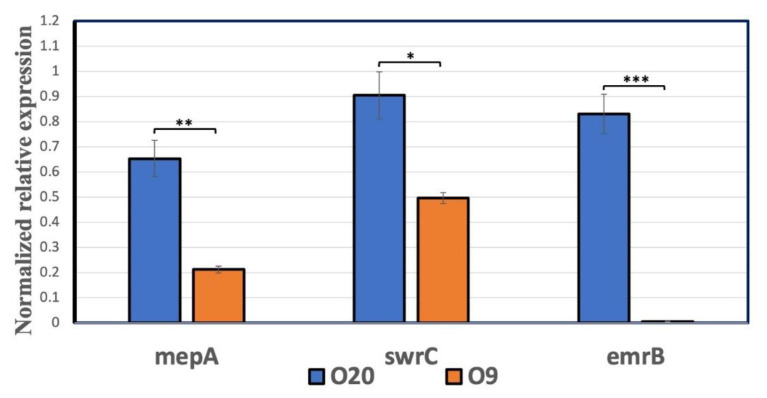
Expression of *mepA*, *swrC* and *emrB* mRNA in *Staphylococcus aureus* O9 and O20 bacterial strains. * Mean significant differences between groups (Unpaired *t*-test, * = *p* < 0.05, ** = *p* < 0.01, *** = *p* < 0.001) and bars represent Means ± SEM (*n* = 3).

**Table 1 antibiotics-11-00759-t001:** Primers for qPCR for different genes identified in the proteomic analysis.

Gene Size (bp)	Sense	(5′–3′)
mepA 132	Forward:	CGCGATTGCAAGTTATGGTATC
	Reverse:	ACGTCTTTCATACGGCCTTTA
swrC 100	Forward:	CAACGCTGGAACGAGTGTAA
	Reverse:	ATCAGGTTCAGTAGGCGAAATG
Emrb 105	Forward	CCACCATATTGCGATGCTAATTC
	Reverse	GACAACGCAAACTACACAACAT
gyraseA 110	Forward:	CAGAGCTCGTTCGTGACAAG
	Reverse:	AGCATTTGCATCCTTACGCA

**Table 2 antibiotics-11-00759-t002:** Proteins identified by LC-MS/MS with transportation functions that were upregulated or unique in *S. aureus* strain O20.

Protein	Gene Name	Fold Change/Unique	Protein Function
A0A0D3Q904	*ecsA3*	1.65	ABC transporter, ATP-binding protein EcsA
A0A0D6GCY7	*mepA*	2.21	MATE family efflux transporter (Multi antimicrobial extrusion protein (Na(+)/drug antiporter)
A0A0D6GDE2	*ytrB1*	2.29	ABC transporter ATP-binding protein
A0A0D6GZT9	*msbA2*	unique	ABC transporter ATP-binding protein
A0A0D6H6R4	*emrB4*	unique	MFS transporter (Multidrug resistance transporter)
A0A0Z0Q5C1	*norB3*	unique	Permease
Q9RQG6	*swrC*	1.70	RND (Resistance nodulation division multidrug transporter)
X5DZI9	*BTN44_13755*	unique	MFS transporter (Major facilitator family transporter)

## Data Availability

All data supporting the findings of the study are present in the manuscript.

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
