# Peer review of "Proteomic Comparison of Ivermectin Sensitive and Resistant Staphylococcus aureus Clinical Isolates Reveals Key Efflux Pumps as Possible Resistance Determinants"

_antibiotics, 2022, doi:10.3390/antibiotics11060759_

Round 1

Reviewer 1 Report

I would like to thank the authors for preparing a revised manuscript. There are two questions, which remain to be answered:

Section 2.4. It is still not clear what part of the cells is isolated. The authors state that "Debris was removed by centrifugation at 4 ËšC for 10 min at 21,000× g." This means that membrane fractions are removed, which is central for this work as efflux pumps are membrane bound. I have to insist on a clear statement here.

The other point is that the authors write that they modified the legend to Figure 4: "We have modified the figure legend accordingly as follows “Figure 4. Pie charts of the functional annotation of all proteins identified in strain O20 (A) and strain O9 (B). GO terms assigned are associated with molecular function from UNIPROT-GOA. Metal and nucleotide-binding proteins are a subdivision of catalytic activity.” The modification does not seem to appear in the revised manuscript.

Author Response

Reviewer comments

We thank the reviewers for their encouraging comments and have now made necessary modifications in the revised manuscript along with a point-by-point response.

Reviewer 1

I would like to thank the authors for preparing a revised manuscript. There are two questions, which remain to be answered:

Question

Section 2.4. It is still not clear what part of the cells is isolated. The authors state that "Debris was removed by centrifugation at 4 ËšC for 10 min at 21,000× g." This means that membrane fractions are removed, which is central for this work as efflux pumps are membrane bound. I have to insist on a clear statement here.

Response

We thank the reviewer on pointing this out and have modified the statement as follows in the revised manuscript.

“Whole proteome analysis was performed as described below. Bacterial cell lysis, protein extraction digestion, reduction, and alkylation were carried out as previously described [20]. Briefly, bacterial cells were harvested by centrifugation at 4ËšC for 15 min at 21,000× g. The resulting pellets (100 mg for each sample) were then solubilized in a buffer (100 mMTris/4% SDS, pH 8). To improve cell lysis, the samples were sonicated with pulses of 10s and 20s on/off for a total of 5 min in cold water to avoid heating. Next, the samples were centrifuged at 4ËšC for 10 min at 21,000×g to remove insoluble material. The supernatant containing proteins was reduced with 20 mM TCEP (Tris(2-carboxyethyl) phosphine) (Sigma-Aldrich, Canada). Non-protein contaminants were removed by 20% trichloroacetic acid/acetone precipitation. Proteins in the precipitate were resolubilized by adding 8 M urea in Tris-HCl pH 8.0 for 30 min at room temperature. To avoid trypsin inhibition in the following steps, the urea concentration was reduced to 2 M urea. Protein reduction was obtained by adding 20 mM TCEP to the sample for 30 min at room temperature. Protein alkylation was achieved by adding 20 mM iodoacetamide to the sample and incubation at room temperature in the dark for 30 min. Protein concentration was determined using a BCA protein assay kit (Thermo Fisher Scientific, Waltham, MA). Proteins (1-2 mg) were digested with modified sequencing grade trypsin (Promega, Madison, WI, USA), for 12 h at 37ËšC. Then, an acidic solution (200 mM NaCl, 0.1% formic acid) was added to stop trypsin reaction. Trypsin and undigested proteins were removed by a 30 kDa MWCO Amicon ultra-centrifugal filter (Millipore Sigma) [24]. The filtrate containing the digested proteins (<30 kDA) was used for further analysis and the peptides were desalted by a centrifugal column (Sep-Pak Plus C-18, Waters Limited, Mississauga, ON) before peptide quantification (Pierce Quantitative Colorimetric Peptide Assay, Thermo-Fisher Scientific, San Jose, CA) and stored at −80 ËšC.”

Question

The other point is that the authors write that they modified the legend to Figure 4: "We have modified the figure legend accordingly as follows “Figure 4. Pie charts of the functional annotation of all proteins identified in strain O20 (A) and strain O9 (B). GO terms assigned are associated with molecular function from UNIPROT-GOA. Metal and nucleotide-binding proteins are a subdivision of catalytic activity.” The modification does not seem to appear in the revised manuscript.

Response

We thank the reviewer for pointing this out, we have now made the amendments in the revised manuscript and added the following statement in the figure legend “Metal and nucleotide-binding proteins are a subdivision of catalytic activity”.

Reviewer 2 Report

The study presented by the authors is very complete since it deals with the efficacy of Ivermectin over Staphylococcus aureus and also analyzes the proteins involved in resistance to the antibiotic. All this is characterized from a genetic and biochemical point of view and is supported by databases of previous studies carried out.

Despite liking the article, I propose changes that can improve the presentation of the study

  • At the section “1. Bacterial Isolates and Resistance Classification “. The authors have reflected:

“Thus, for this study we arbitrarily defined the sensitive strain as the one which could be killed by IVM at the concentration of <12.5 μg/ml, while the resistant strain was not killed by IVM up to 100 μg/ml concentration”

Why? What are they based on? Perhaps this should be reflected only in results.

  • Also, they wrote “To increase the chances for identification and quantitation of peptides, in the LC- MS/MS analysis, a total of 12 unique gradients of the LC solvents were applied. Authors should reflect gradients in methodology--- what gradients?
  • The experiments to know the effects of IVM on Bacterial Growth, were repeated?
  • The size of the genes amplified by PCR have not been reflected
  • The description of Supplementary Materials remains to be done, I think it has not been reflected in the main text of the article
  • In discussion, no reference was made to the figures that support the results
  • And, Why are there paragraphs in Yellow?

Author Response

Reviewer 2

The study presented by the authors is very complete since it deals with the efficacy of Ivermectin over Staphylococcus aureus and also analyzes the proteins involved in resistance to the antibiotic. All this is characterized from a genetic and biochemical point of view and is supported by databases of previous studies carried out.

Response

We thank the reviewer for the nice comments

Question

Despite liking the article, I propose changes that can improve the presentation of the study

At the section “1. Bacterial Isolates and Resistance Classification “. The authors have reflected:

“Thus, for this study we arbitrarily defined the sensitive strain as the one which could be killed by IVM at the concentration of >12.5 μg/ml, while the resistant strain was not killed by IVM up to 100 μg/ml concentration”

Why? What are they based on? Perhaps this should be reflected only in results.

Response

We thank the reviewer on pointing this out. However, the protocols for establishing the sensitivities of IVM against S. aureus are not established, thus, as stated in the text we have proposed this statement for the current study. We have also included an additional statement in the results section of the revised manuscript to further emphasize the scenario, as below.

“Notably, as stated above since there are no established CLSI protocols for prediction of IVM resistance against S. aureus, hence for this study we have defined the sensitive and tolerant IVM strains as the ones which could be killed/survive in the presence of IVM at the concentrations of >12.5 or up to 100 μg/ml, respectively.”

Question

Also, they wrote “To increase the chances for identification and quantitation of peptides, in the LC- MS/MS analysis, a total of 12 unique gradients of the LC solvents were applied. Authors should reflect gradients in methodology--- what gradients?

Response

We thank the reviewer for pointing this out and have modified the statement as follows in the revised manuscript

“To generate a peptide fractionation, 12 unique gradients of the LC solvents were ap-plied to the column for 90 minutes each. The gradients consisted of a mixture of buffer A (H2O, 0.1 % formic acid) and buffer B (CH3CN, 0.1% formic acid). Each gradient consisted in the following concentrations of buffer B: 0%, 5%, 7.5%, 10%, 12.5%, 15%, 17.5%, 20%, 25%, 35%, 50%, 100%, in combination with buffer A”

Question

The experiments to know the effects of IVM on Bacterial Growth, were repeated?

Response

Yes, the experiments investigating the effects of IVM on the growth of bacteria were repeated 3 times with 3 independent replicates, n=9. We have now added this information in the revised manuscript.

Question

The size of the genes amplified by PCR have not been reflected

Response

We thank the reviewer for pointing this out and have now included the size of the qPCR amplified genes.

Question

The description of Supplementary Materials remains to be done, I think it has not been reflected in the main text of the article

Response

The following description was added to the Supplementary Materials section:

Supplementary Materials:

Protein Table – A table format file containing the proteins detected and quantified in S. aureus strain O9 and O20.

Heatmap_SA.html – The interactive version of the heatmap (Figure 6) generated from the Proteomics analysis of S. aureus strain O9 and O20.

Also, in the main text, page 9, section 3.2 Proteomic analysis, the following text is present:

A list of all proteins found in this study, quantified and normalized is presented in the Supplementary materials (Protein Table).

On page 10 of the same section:

For a detailed view of Figure 6, an interactive version can be found in the file “Heatmap_SA.html”, in theSupplementary materials section.

Question

In discussion, no reference was made to the figures that support the results

Response

We have now referred to the figures in the discussion that support the results.

Question

And, Why are there paragraphs in Yellow?

Response

The yellow text was the new inclusions added as per the request of the academic editor. However, we have now removed all the highlighted yellow part and turned track changes on for any additions in the revised manuscript.

This manuscript is a resubmission of an earlier submission. The following is a list of the peer review reports and author responses from that submission.

Round 1

Reviewer 1 Report

The present manuscript investigates Ivermectin’s (IVM) resistance of two Staphylococcus aureus isolates (O9 and O20) using proteomics and gene expression analysis. The authors analysed IVM effects on both strains using growth curves and time-kill kinetics and therefore identified one strain as IVM-sensitive (O9) and the other one as IVM-resistant (O20). The authors also focused their analysis on proteomic differences between both strains and found an overexpression of some efflux pumps in the IVM-resistant strain. Drug efflux has then been proposed as a potential mechanism involved in IVM resistance.

This study provides valuable results that could be strengthen by addressing some points:

  • As mentioned by the authors, a comparison with a standard strain of S. aureus is missing, this could have been useful to emphasize resistance or the sensitivity of both isolates used in this study. It is a control that should be added.
  • As mentioned in the manuscript, the MRSA (methicillin resistant Staphylococcus aureus) strains are prevalent throughout the globe and they  are  often  resistant  to  most  of  the  known antibiocs. It would have been useful to check whether the O9 and O20 isolates could be MRSA strains and to investigate if there is any cross-resistance mechanisms between methicillin and Ivermectin resistances. Mainly because there is a pump (EmrB) that has been identified in this study as a potential actor of IVM-resistance. EmrB is involved in β-lactams, precisely ampicillin resistance in S. aureus. Methicillin resistance results from a decreased affinity for most β-lactam antibiotics in MRSA strains due to the production of an altered penicillin-binding protein known as PBP2a. As Methicillin is also a β-lactam, it could bring more insights to investigate and compare methicillin and Ivermectin resistances to check if they share any similar features.
  • DMSO is used in the study as a solvent control, I guess it is because Ivermectin has been prepared using it as a solvent. But in the growth curves and the time-kill kinetics, a curve without IVM nor DMSO as a control is missing.
  • The authors proposed drug efflux as a potential mechanism involved in IVM resistance. But, to prepare the peptides for LC MS/MS analysis, the authors have digested their proteins extract and used a 30 kDa MWCO Amicon ultra-centrifugal filter to remove trypsin and undigested proteins. As a peptide resulting from the digestion might be smaller than the protein it comes from, I am wondering if by using a filtration method with such a cutoff, the authors may have lost some proteins of interest involving another mechanism in their investigation of IVM resistance

Author Response

Reviewer 1 comments

The present manuscript investigates Ivermectin’s (IVM) resistance of two Staphylococcus aureusisolates (O9 and O20) using proteomics and gene expression analysis. The authors analysed IVM effects on both strains using growth curves and time-kill kinetics and therefore identified one strain as IVM-sensitive (O9) and the other one as IVM-resistant (O20). The authors also focused their analysis on proteomic differences between both strains and found an overexpression of some efflux pumps in the IVM-resistant strain. Drug efflux has then been proposed as a potential mechanism involved in IVM resistance.

Response

We thank the reviewer for your encouraging comments and we have responded to each point raised below.

This study provides valuable results that could be strengthen by addressing some points:

As mentioned by the authors, a comparison with a standard strain of S. aureus is missing, this could have been useful to emphasize resistance or the sensitivity of both isolates used in this study. It is a control that should be added.

Response

While we agree that a standard ATCC strain could be added to compare the sensitivity patterns to Ivermectin, we did not perform this for the revision. Due to COVID-19, our laboratories are closed. Hence, this additional experiment cannot be performed at this  time. , and we would like to publish our findings as soon as possible for the larger interest of the scientific community. As acknowledged by the reviewer we have thus included this a potential limitation of the study in line 343.

As mentioned in the manuscript, the MRSA (methicillin resistant Staphylococcus aureus) strains are prevalent throughout the globe and they  are  often  resistant  to  most  of  the  known antibiocs. It would have been useful to check whether the O9 and O20 isolates could be MRSA strains and to investigate if there is any cross-resistance mechanisms between methicillin and Ivermectin resistances. Mainly because there is a pump (EmrB) that has been identified in this study as a potential actor of IVM-resistance. EmrB is involved in β-lactams, precisely ampicillin resistance in S. aureus. Methicillin resistance results from a decreased affinity for most β-lactam antibiotics in MRSA strains due to the production of an altered penicillin-binding protein known as PBP2a. As Methicillin is also a β-lactam, it could bring more insights to investigate and compare methicillin and Ivermectin resistances to check if they share any similar features.

Response

These two strains are sensitive to methicillin as well as cefoxitin and are thus classified as methicillin sensitive staphylococcus aureus (MSSA). We have now added this information in the revised manuscript.

DMSO is used in the study as a solvent control, I guess it is because Ivermectin has been prepared using it as a solvent. But in the growth curves and the time-kill kinetics, a curve without IVM nor DMSO as a control is missing.

Response

Yes, DMSO was used as a solvent control. As shown in the Figure 1, it does not have an inhibitory or stimulatory effect on the growth of Staphylococcus aureus. For this reason, we did not add it to the graphs. However, upon the suggestion of the reviewer we have now added an additional curve and named it the control, please see below

The authors proposed drug efflux as a potential mechanism involved in IVM resistance. But, to prepare the peptides for LC MS/MS analysis, the authors have digested their proteins extract and used a 30 kDa MWCO Amicon ultra-centrifugal filter to remove trypsin and undigested proteins. As a peptide resulting from the digestion might be smaller than the protein it comes from, I am wondering if by using a filtration method with such a cutoff, the authors may have lost some proteins of interest involving another mechanism in their investigation of IVM resistance

Response

The reviewer points out an important aspect of the sample preparation method for proteomics. We would like to clarify that once the proteins are digested with Trypsin, most of the proteins are partially or completely digested to smaller peptides (~ 20kDa). We filtered the sample after the digestion in order to remove undigested proteins or large peptides that would not be completely ionized, making its detection less efficient, in the mass spectrometer.

This approach is based on the Filter-aided sample preparation (FASP) method (J.R. Wisniewski, A. Zougman, N. Nagaraj, M. Mann. Universal sample preparation method for proteome analysis. Nature Methods, 6 (5) (2009), pp. 359-362, 10.1038/nmeth.1322).

However, we modified the above protocol to allow for the preparation of a higher protein concentration.

Reviewer 2 Report

The manuscript by Shoaib Ashraf et al describes the proteomic differences between an ivermectin-sensitive and -resistant S. aureus isolate. They found unique proteins in each of the isolates followed by differentially expressed transport proteins. Among these, they report three efflux pumps that show overexpression in the resistant S. aureus isolate. The authors propose the overexpression of these efflux pumps as possible drivers of observed resistance to ivermectin in S. aureus. Overall, the manuscript is well written and interesting to read. My comments that might help to further improve the manuscript are as follows:

In the title, should ‘Proteomics’ be ‘Proteomic’? Please check that proteomics and proteomic are used correctly throughout.

Line 19 remove the extra period at the end of the sentence

Lines 33 “organism” should be “microorganism”. Check it throughout the text.

Line 55 It’s not clear what is meant by “resilient” here; resistant or Multiresistant?

Lines 57-60 maybe it would be better to move these criteria to the method or result section.

Lines 72-75 it might be good and more relevant to add a recent reference highlighting the quest for inhibition of efflux pump mediated antimicrobial resistance in staphylococcus

Line 79 make it a bit specific for the reader by stating maybe a number or percentage rather than “many”

In the method section 2.2, it might be good to add about your biological/technical replicates and whether you used 96 well trays and microplate reader for OD values?

Line 99 “experiments” would be preferable instead of “results” as you have not yet presented the results.

Line 103 it would be good to include the corresponding μg/ml of the stated MIC range for the reader.

In the method section, all “g” for centrifugations should be “x g”. e.g line 112, centrifugation … at “21,000 x g” rather than “21,000 g”

Line 132, “60” at the start of the sentence should be as written as “Sixty”

Fig 1, should DMSO curve in Fig 1A be in yellow like in other panels of the figure?

In Fig 1, I recommend adding error bars for A and B as shown in D and C.  

Fig 3, I am not sure if the colors in the graph bar relate to the Venn diagram? I assume that Fig 3A shows numbers of unchanged, upregulated, and downregulated proteins in the resistant isolate alone? The 326 unique proteins in strain O20 as shown in Fig 3B is the sum of downregulated and upregulated bars in Fig 3A? It would be good if you clarify the figure to help the reader follow the link between the graph and the Venn diagram.

Line 240, are you referring to two tables in the supplementary material? It seems there is only one supplementary table.   

Table 2 states that swrC is an uncharacterized protein. We can find in the literature that swrC is an RND multidrug efflux transporter.  

Line 262, the “,” after S. aureus should be “.”

Line 293, I think it would be better to move this sentence earlier, perhaps in the last part of the introduction.

Lin 316 “isolates” should be “isolate”

It will be interesting to provide an explanation somewhere in the discussion as to why the downregulation of unique proteins in the resistant isolate has not been considered when investigating the resistance mechanism.   

Author Response

Reviewer 2 comments

The manuscript by Shoaib Ashraf et al describes the proteomic differences between an ivermectin-sensitive and -resistant S. aureus isolate. They found unique proteins in each of the isolates followed by differentially expressed transport proteins. Among these, they report three efflux pumps that show overexpression in the resistant S. aureus isolate. The authors propose the overexpression of these efflux pumps as possible drivers of observed resistance to ivermectin in S. aureus. Overall, the manuscript is well written and interesting to read. My comments that might help to further improve the manuscript are as follows:

Response

We thank the reviewer for your encouraging comments and we have responded to each point raised below.

In the title, should ‘Proteomics’ be ‘Proteomic’? Please check that proteomics and proteomic are used correctly throughout.

Response

Proteomics in the title and throughout the manuscript have been replaced with proteomic, when it is suitable.

Line 19 remove the extra period at the end of the sentence

Response

Extra period has been removed.

Lines 33 “organism” should be “microorganism”. Check it throughout the text.

Response

Organism has been replaced with microorganism throughout the text.

Line 55 It’s not clear what is meant by “resilient” here; resistant or Multiresistant?

Response

Resilient has been replaced by resistant (now on line 56).

Lines 57-60 maybe it would be better to move these criteria to the method or result section.

Response

The criteria have been moved to the methods section

Lines 72-75 it might be good and more relevant to add a recent reference highlighting the quest for inhibition of efflux pump mediated antimicrobial resistance in staphylococcus

Response

The reference has been added where efflux pump inhibitors have been used to overcome antibiotic resistance in Staphylococcus species. The text is follows “To this end, different efflux pump inhibitors have been explored as a strategy to overcome antibiotic resistance in S. aureus species [19]”

[19]. Dashtbani-Roozbehani, A.; Brown, M.H. Efflux pump mediated antimicrobial resistance by Staphylococci in health-related environments: Challenges and the quest for inhibition. Antibiotics 2021, 10, 1502.

Line 79 make it a bit specific for the reader by stating maybe a number or percentage rather than “many”

Response

We have added the percentage and have replaced many with “ approximately 10 percent” (line 84).

In the method section 2.2, it might be good to add about your biological/technical replicates and whether you used 96 well trays and microplate reader for OD values?

Response

We have now added the biological/technical replicates in the manuscript method section 2.2 as follows “Three biological and technical replicates were performed for each experiment throughout the study.”

Line 99 “experiments” would be preferable instead of “results” as you have not yet presented the results.

Response

Results were replaced with experiments (line 114).

Line 103 it would be good to include the corresponding μg/ml of the stated MIC range for the reader.

Response

We have now included the corresponding μg/ml of the stated MIC as follows “Following this, IVM was added at concentrations of 12X(MIC) (150 μg/ml), 8X(MIC) (100 μg/ml), 4X(MIC) (50 μg/ml), 2X(MIC) (25 μg/ml), 1X(MIC) (12.5 μg/ml), 1/2X(MIC) (6.25 μg/ml), 1/4X(MIC) (3.125 μg/ml), and 0X(MIC) (1.562 μg/ml)”.

In the method section, all “g” for centrifugations should be “x g”. e.g line 112, centrifugation … at “21,000 x g” rather than “21,000 g”

Response

We have now made the change.

Line 132, “60” at the start of the sentence should be as written as “Sixty”

Response

60 has been replaced with sixty at the start of the sentence.

Fig 1, should DMSO curve in Fig 1A be in yellow like in other panels of the figure?

Response

The DMSO curve in Fig1A has been made yellow now

In Fig 1, I recommend adding error bars for A and B as shown in D and C.  

Response

Error bars have been added for panels A and B of Fig 1 (above).

Fig 3, I am not sure if the colors in the graph bar relate to the Venn diagram? I assume that Fig 3A shows numbers of unchanged, upregulated, and downregulated proteins in the resistant isolate alone? The 326 unique proteins in strain O20 as shown in Fig 3B is the sum of downregulated and upregulated bars in Fig 3A? It would be good if you clarify the figure to help the reader follow the link between the graph and the Venn diagram.

Response

The original text was:

“Fig 3A) shows the distribution of the protein classifications (upregulated, downregulated or unchanged) rom sample O20 when compared to O9. Figure 3B) shows the Venn diagram of both samples (O20 and O9).”

The text has been modified to:

“Figure 3. Differentially abundant and unique proteins found in strains O9 and O20. A) Distribution of the proteins found in strain O20 that were found to be unchanged, upregulated, and downregulated compared to the strain O9 of Staphylococcus aureus. The x-axis shows the number of proteins. B) The Venn diagram of proteins identified in the strains O9 and O20.  

Line 240, are you referring to two tables in the supplementary material? It seems there is only one supplementary table.   

 Response

The “Table 2” in the text refers to the “Table 2. Proteins identified by LC-MS/MS with transportation functions that are upregulated or unique in S. aureus strain O20. However, one supplementary file was missing which has now been provided as an additional file.

Table 2 states that swrC is an uncharacterized protein. We can find in the literature that swrC is an RND multidrug efflux transporter.  

 Response

We have fixed this error and modified the table 2 accordingly as below.

Protein

Gene name

Fold change/unique

Protein function

A0A0D3Q904

ecsA_3

unique

ABC transporter, ATP-binding protein EcsA

A0A0D6GCY7

mepA

2.21

MATE family efflux transporter (Multi antimicrobial extrusion protein (Na(+)/drug antiporter)

A0A0D6GDE2

ytrB_1

unique

ABC transporter ATP-binding protein

A0A0D6GZT9

msbA_2

unique

ABC transporter ATP-binding protein

A0A0D6H6R4

emrB_4

unique

MFS transporter (Multidrug resistance transporter)

A0A0Z0Q5C1

norB_3

unique

Permease

Q9RQG6

swrC

1.70

RND (Resistance nodulation division multidrug transporter)

X5DZI9

BTN44_13755

unique

MFS transporter (Major facilitator family transporter)

Line 262, the “,” after S. aureus should be “.”

Response

We have now made the “,” to “.”

Line 293, I think it would be better to move this sentence earlier, perhaps in the last part of the introduction.

Response

The sentence has been moved to the last part of the introduction.

Lin 316 “isolates” should be “isolate”

 Response

Isolates have been replaced with isolate.

It will be interesting to provide an explanation somewhere in the discussion as to why the downregulation of unique proteins in the resistant isolate has not been considered when investigating the resistance mechanism.   

Response

The downregulated proteins in the resistant isolate were also considered in the pool of proteins that we analyzed. Our approach was to look specifically for transporters, since they present an important role in drug resistance, and in this dataset, they were either unique or upregulated.

Reviewer 3 Report

General:

The authors describe a comparison of two Staphylococcus aureus isolates with different sensitivity to ivermectin in vitro. Discussion and interpretation of the results focus on efflux pumps. The topic is of high importance in the light of MRSA infections and of antibiotic resistance in general.

Broad comments:

  1. It would be important to explain the selection of the two isolates used in the study in more detail. How many isolates were tested on ivermectin sensitivity and how do the sensitivities of O9 and O20 compare to the overall distribution? What are the sensitivities of O9 and O20 against classes of antibiotics with known mode of action?

  1. The evaluation and discussion of the data concentrates on efflux pumps. It would be highly interesting to know, which other proteins were differently expressed in the two strains (up- and down-regulated). I suggest to add more information on the results and to discuss the findings to speculate about resistance mechanism in a broader sense. A broad analysis might also allow speculation on the mechanism of action of ivermectin.

  1. It should be shown that overexpression of the identified efflux pumps is relevant. The authors lists a number of compounds, known to be substrates of the three efflux pumps in the discussion. Please provide susceptibility data to show if sensitivity towards substrates of the pumps are different in the two isolates.

Specific comments:

p1, lines 30 ff. Significance and Introduction. Please clearly state for which indications ivermectin is approved and what are preclinical activities.

p1, lines 30. Did ivermectin win the noble price or was it the scientist who discovered ivermectin?

p2, lines 56-59. This part should be moved to the methods section and needs to be justified. Why were these cutoff values applied (see also broad comment 1).

p2, lines 67-69. Is this true for S. aureus in general or does the WHO (and others) list MRSA?

p3, lines 93-94. Should the DMSO content not be kept constant to avoid unspecific effects?

Section 2.4. Please add a statement to explain which proteins were isolated (whole proteome or soluble part?).

p4, line 142. “used” or similar instead of “inputted”?

Figure 1. Please describe the panels in alphabetic order (A, C before B, D).

p7, lines 217-218. It is not clear what the authors want to say here.

p7, line 218. How does the total number of 2165 identified proteins relate to the total number of proteins in S. aureus?

Figure 3. What is the unit of the numbers below the bars?

Figure 4. Is it correct that DNA binding and RNA binding proteins are shown in different color codes for O9 and O20? If yes, I suggest using the same colors in both graphs.

Figure 4. What proteins are included in the category “catalytic activity”? Metal- and nucleotide-binding proteins often have a catalytic activity.

Table 2. Did you identify differently downregulated transporters? Does unique mean “only detected in O20” or are these systems truly absent in O9?

Author Response

Reviewer 3 comments

General:

The authors describe a comparison of two Staphylococcus aureus isolates with different sensitivity to ivermectin in vitro. Discussion and interpretation of the results focus on efflux pumps. The topic is of high importance in the light of MRSA infections and of antibiotic resistance in general.

 Response

We thank the reviewer for your encouraging comments and we have responded to each point raised below.

Broad comments:

It would be important to explain the selection of the two isolates used in the study in more detail. How many isolates were tested on ivermectin sensitivity and how do the sensitivities of O9 and O20 compare to the overall distribution? What are the sensitivities of O9 and O20 against classes of antibiotics with known mode of action?

 Response

The following text has been added in the materials and methods section of the revised manuscript

Two S. aureus clinical isolates (O9 and O20) were used in the present study. The isolates were isolated and identified as S. aureus from mastitis cases from Lahore, Pakistan. Following initial isolation and identification, the isolates were shipped to McGill University, Canada for further analyses. Both strains were sensitive to methicillin, and cefoxitin classifying them as methicillin sensitive Staphylococcus aureus (MSSA). The bacteria were grown in Cation-adjusted Mueller Hinton (MH) broth and tryptic soya broth (TSB) (Sigma Aldrich, Canada) at 37°C. There are no standard guidelines developed by the Clinical and Laboratory Standard Institute (CLSI) for the use of IVM against bacteria. Thus, for this study we define the sensitive strain as the one which could be killed by IVM at the concentration of <12 μg/ml versus the resistant strain which was not killed by IVM upto 100 μg/ml concentration.

The evaluation and discussion of the data concentrates on efflux pumps. It would be highly interesting to know, which other proteins were differently expressed in the two strains (up- and down-regulated). I suggest to add more information on the results and to discuss the findings to speculate about resistance mechanism in a broader sense. A broad analysis might also allow speculation on the mechanism of action of ivermectin. It should be shown that overexpression of the identified efflux pumps is relevant. The authors lists a number of compounds, known to be substrates of the three efflux pumps in the discussion. Please provide susceptibility data to show if sensitivity towards substrates of the pumps are different in the two isolates.

Response

Both up and downregulated proteins in the resistant isolate were considered in the proteomic analysis. We specifically targeted the transporters, since they present an important role in drug resistance, and in this dataset, they were either unique or upregulated. While we agree with the reviewer that in order to comprehend the exact relationship between efflux pumps, IVM and S. aureus, it is a must to block these transporters with known blocking against that would reverse the resistant phenotype. However, due to the pandemic the laboratories are closed, and it is impossible for us to carry out these experiments at this point of time.

Specific comments:

p1, lines 30 ff. Significance and Introduction. Please clearly state for which indications ivermectin is approved and what are preclinical activities.

Response

Relevant text added in the introduction.

p1, lines 30. Did ivermectin win the noble price or was it the scientist who discovered ivermectin?

Response

We have now modified the statement as follows “Ivermectin (IVM) is popular as a miracle drug and the scientists who discovered the drug were awarded the Noble Prize in Physiology and Medicine for the year 2015.”

p2, lines 56-59. This part should be moved to the methods section and needs to be justified. Why were these cutoff values applied (see also broad comment 1).

Response

The part has been moved to materials and methods section

p2, lines 67-69. Is this true for S. aureus in general or does the WHO (and others) list MRSA?

Response

The WHO classifies both S. aureus as well as MRSA in the list of bacteria that are of high priority and require new antibiotics, please see link below

https://www.who.int/news/item/27-02-2017-who-publishes-list-of-bacteria-for-which-new-antibiotics-are-urgently-needed

p3, lines 93-94. Should the DMSO content not be kept constant to avoid unspecific effects?

Response

We agree with the reviewer, there was typographical error, so we correct it as follows “IVM dissolved in dimethylsulfoxide (DMSO) (solvent control) was then serially titrated at concentrations of 100, 50, 25, 12.5, 6.25, 3.125 and 1.56 μg/ml.”

Section 2.4. Please add a statement to explain which proteins were isolated (whole proteome or soluble part?).

Response

The following statement was added: “Protein extract of cell pellets (100 mg) from each bacterial strain was digested as previously described [21] and whole proteome analysis was carried out as follows”

p4, line 142. “used” or similar instead of “inputted”?

Response

Inputted is replaced with “used”.

Figure 1. Please describe the panels in alphabetic order (A, C before B, D).

Response

The panels are now described in the alphabetic order.

p7, lines 217-218. It is not clear what the authors want to say here.

Response

We modified the text of the methods section and the text in the item 3.2 for clarification.

p7, line 218. How does the total number of 2165 identified proteins relate to the total number of proteins in S. aureus?

Response

The following sentence has been added: “A total of 2,165 proteins were identified in the dataset of strains O9 and O20, representing about 75% of the proteins from the reference proteome in UNIPROT database for S. aureus strain NCTC/PS47 (proteome ID: UP000008816).”

Figure 3. What is the unit of the numbers below the bars?

Response

The figure legend has been modified to: “Figure 3. Differentially abundant and unique proteins found in strains O9 and O20. A) Distribution of the proteins found in strain O20 that were found to be unchanged, upregulated, and downregulated compared to the strain O9 of Staphylococcus aureus. The x-axis shows the number of proteins. B) The Venn diagram of proteins identified in the strains O9 and O20.”

Figure 4. Is it correct that DNA binding and RNA binding proteins are shown in different color codes for O9 and O20? If yes, I suggest using the same colors in both graphs.

Response

We modified the figure as suggested by the reviewer, please see below.

Figure 4. What proteins are included in the category “catalytic activity”? Metal- and nucleotide-binding proteins often have a catalytic activity.

Response

Metal and nucleotide-binding proteins are a subdivision of catalytic activity.

Table 2. Did you identify differently downregulated transporters? Does unique mean “only detected in O20” or are these systems truly absent in O9?

Response

We use the term unique, as it is mostly used in proteomics, to report that the protein was only identified in the specific sample (in this case, in the O20). It does not necessarily mean that the protein is not expressed in the other sample, but rather that the protein was not detected by the mass spectrometer in the other sample, which can happen due to the intrinsic limitations of MS-based proteomics, such as low signal of peptides. In our study, we limited this issue by analyzing a high amount/concentration of proteins, and by highly fractionating the sample in a MudPIT strategy, as explained in our methodology.

Round 2

Reviewer 1 Report

The authors have addressed most of the concerns mentioned above. However,

1- The citation mentioned below is missing in the manuscript.

The authors proposed drug efflux as a potential mechanism involved in IVM resistance. But, to prepare the peptides for LC MS/MS analysis, the authors have digested their proteins extract and used a 30 kDa MWCO Amicon ultra-centrifugal filter to remove trypsin and undigested proteins. As a peptide resulting from the digestion might be smaller than the protein it comes from, I am wondering if by using a filtration method with such a cutoff, the authors may have lost some proteins of interest involving another mechanism in their investigation of IVM resistance

Response

The reviewer points out an important aspect of the sample preparation method for proteomics. We would like to clarify that once the proteins are digested with Trypsin, most of the proteins are partially or completely digested to smaller peptides (~ 20kDa). We filtered the sample after the digestion in order to remove undigested proteins or large peptides that would not be completely ionized, making its detection less efficient, in the mass spectrometer.

This approach is based on the Filter-aided sample preparation (FASP) method (J.R. Wisniewski, A. Zougman, N. Nagaraj, M. Mann. Universal sample preparation method for proteome analysis. Nature Methods, 6 (5) (2009), pp. 359-362, 10.1038/nmeth.1322).

2- From the answer above, the authors should therefore make clear in the materials and methods section that the peptides used for mass spectrometry analysis are recovered from the filtrate, and maybe also add the peptides size after digestion, because the way it is written makes it confusing.

3- I understand the COVID restrictions the authors have to face and I also understand that this affects the manuscript revision, but I keep thinking that having more controls will strengthen the results presented in this study.

Author Response

Reviewer 1

The authors have addressed most of the concerns mentioned above. However,

Response

We thank the reviewer for encouraging comments and below try to address each point raised

1- The citation mentioned below is missing in the manuscript.

The authors proposed drug efflux as a potential mechanism involved in IVM resistance. But, to prepare the peptides for LC MS/MS analysis, the authors have digested their proteins extract and used a 30 kDa MWCO Amicon ultra-centrifugal filter to remove trypsin and undigested proteins. As a peptide resulting from the digestion might be smaller than the protein it comes from, I am wondering if by using a filtration method with such a cutoff, the authors may have lost some proteins of interest involving another mechanism in their investigation of IVM resistance

 Response

The reviewer points out an important aspect of the sample preparation method for proteomics. We would like to clarify that once the proteins are digested with Trypsin, most of the proteins are partially or completely digested to smaller peptides (~ 20kDa). We filtered the sample after the digestion in order to remove undigested proteins or large peptides that would not be completely ionized, making its detection less efficient, in the mass spectrometer.

This approach is based on the Filter-aided sample preparation (FASP) method (J.R. Wisniewski, A. Zougman, N. Nagaraj, M. Mann. Universal sample preparation method for proteome analysis. Nature Methods, 6 (5) (2009), pp. 359-362, 10.1038/nmeth.1322).

Response

We have added the reference in the revision, as stated in the response to the next question as follows “Trypsin and undigested proteins were removed by a 30 kDa MWCO Amicon ultra-centrifugal filter (Millipore Sigma) [24].”

2- From the answer above, the authors should therefore make clear in the materials and methods section that the peptides used for mass spectrometry analysis are recovered from the filtrate, and maybe also add the peptides size after digestion, because the way it is written makes it confusing.

Response

We have now included the citation in the manuscript and revised it as follows “Trypsin and undigested proteins were removed by a 30 kDa MWCO Amicon ultra-centrifugal filter (Millipore Sigma) [24]. The filtrate containing the digested proteins (<30 kDA) was used for further analysis and the peptides were desalted by a centrifugal column (Sep-Pak Plus C-18, Waters Limited, Mississauga, ON.)”

3- I understand the COVID restrictions the authors have to face and I also understand that this affects the manuscript revision, but I keep thinking that having more controls will strengthen the results presented in this study.

Response

We thank the reviewer for understanding and acknowledge the importance of controls. In the future experiments, we will include wild type reference strains from CDC.

Reviewer 3 Report

I want to thank the authors for the revised manuscript and the answers to my comments. In some cases, I miss an answer or find it incomplete. Thus, I suggest to add information where indicated below.

General:

The authors describe a comparison of two Staphylococcus aureus isolates with different sensitivity to ivermectin in vitro. Discussion and interpretation of the results focus on efflux pumps. The topic is of high importance in the light of MRSA infections and of antibiotic resistance in general.

Response

We thank the reviewer for your encouraging comments and we have responded to each point raised below.

Broad comments:

It would be important to explain the selection of the two isolates used in the study in more detail. How many isolates were tested on ivermectin sensitivity and how do the sensitivities of O9 and O20 compare to the overall distribution? What are the sensitivities of O9 and O20 against classes of antibiotics with known mode of action?

Response

The following text has been added in the materials and methods section of the revised manuscript

Two S. aureus clinical isolates (O9 and O20) were used in the present study. The isolates were isolated and identified as S. aureus from mastitis cases from Lahore, Pakistan. Following initial isolation and identification, the isolates were shipped to McGill University, Canada for further analyses. Both strains were sensitive to methicillin, and cefoxitin classifying them as methicillin sensitive Staphylococcus aureus (MSSA). The bacteria were grown in Cation-adjusted Mueller Hinton (MH) broth and tryptic soya broth (TSB) (Sigma Aldrich, Canada) at 37°C. There are no standard guidelines developed by the Clinical and Laboratory Standard Institute (CLSI) for the use of IVM against bacteria. Thus, for this study we define the sensitive strain as the one which could be killed by IVM at the concentration of <12 μg/ml versus the resistant strain which was not killed by IVM upto 100 μg/ml concentration.

I was more wondering how the phenotypes of these isolates compare to other mastitis isolates. Please add a statement if IVM resistance is common amongst such isolates or if it is rare. Please add the requested information about resistance/ sensitivity to known classes of antibiotics in addition to methicillin and cefoxitin (especially for agents, which are substrates of the described pumps).

The evaluation and discussion of the data concentrates on efflux pumps. It would be highly interesting to know, which other proteins were differently expressed in the two strains (up- and down-regulated). I suggest to add more information on the results and to discuss the findings to speculate about resistance mechanism in a broader sense. A broad analysis might also allow speculation on the mechanism of action of ivermectin. It should be shown that overexpression of the identified efflux pumps is relevant. The authors lists a number of compounds, known to be substrates of the three efflux pumps in the discussion. Please provide susceptibility data to show if sensitivity towards substrates of the pumps are different in the two isolates.

Response

Both up and downregulated proteins in the resistant isolate were considered in the proteomic analysis. We specifically targeted the transporters, since they present an important role in drug resistance, and in this dataset, they were either unique or upregulated. While we agree with the reviewer that in order to comprehend the exact relationship between efflux pumps, IVM and S. aureus, it is a must to block these transporters with known blocking against that would reverse the resistant phenotype. However, due to the pandemic the laboratories are closed, and it is impossible for us to carry out these experiments at this point of time.

Please show how expression levels of the efflux pumps rank in comparison to other up-regulated or unique proteins. I did not ask for blocking efflux pumps but for evidence that pump substrates other than IVM are affected by the overexpression. This relates to my previous comment on antibiotic resistance patterns.

Specific comments:

p1, lines 30 ff. Significance and Introduction. Please clearly state for which indications ivermectin is approved and what are preclinical activities.

Response

Relevant text added in the introduction.

p1, lines 30. Did ivermectin win the noble price or was it the scientist who discovered ivermectin?

Response

We have now modified the statement as follows “Ivermectin (IVM) is popular as a miracle drug and the scientists who discovered the drug were awarded the Noble Prize in Physiology and Medicine for the year 2015.”

p2, lines 56-59. This part should be moved to the methods section and needs to be justified. Why were these cutoff values applied (see also broad comment 1).

Response

The part has been moved to materials and methods section

p2, lines 67-69. Is this true for S. aureus in general or does the WHO (and others) list MRSA?

Response

The WHO classifies both S. aureus as well as MRSA in the list of bacteria that are of high priority and require new antibiotics, please see link below

https://www.who.int/news/item/27-02-2017-who-publishes-list-of-bacteria-for-which-new-antibiotics-are-urgently-needed

The WHO lists S. aureus as top priority pathogen if it is resistant to methicillin or to vancomycin: “Staphylococcus aureus, methicillin-resistant, vancomycin-intermediate and resistant”. The CDC lists MRSA as a serious threat. Please specify this fact in the manuscript.

p3, lines 93-94. Should the DMSO content not be kept constant to avoid unspecific effects?

Response

We agree with the reviewer, there was typographical error, so we correct it as follows “IVM dissolved in dimethylsulfoxide (DMSO) (solvent control) was then serially titrated at concentrations of 100, 50, 25, 12.5, 6.25, 3.125 and 1.56 μg/ml.”

This procedure apparently leads to a titration of DMSO. Please specify if the final DMSO content was equal at all concentrations.

Section 2.4. Please add a statement to explain which proteins were isolated (whole proteome or soluble part?).

Response

The following statement was added: “Protein extract of cell pellets (100 mg) from each bacterial strain was digested as previously described [21] and whole proteome analysis was carried out as follows”

Please clarify if the sample preparation results in the isolation of soluble proteins only. It is important to understand if efflux pumps are likely to be extracted or not.

p4, line 142. “used” or similar instead of “inputted”?

Response

Inputted is replaced with “used”.

Figure 1. Please describe the panels in alphabetic order (A, C before B, D).

Response

The panels are now described in the alphabetic order.

p7, lines 217-218. It is not clear what the authors want to say here.

Response

We modified the text of the methods section and the text in the item 3.2 for clarification.

p7, line 218. How does the total number of 2165 identified proteins relate to the total number of proteins in S. aureus?

Response

The following sentence has been added: “A total of 2,165 proteins were identified in the dataset of strains O9 and O20, representing about 75% of the proteins from the reference proteome in UNIPROT database for S. aureus strain NCTC/PS47 (proteome ID: UP000008816).”

Figure 3. What is the unit of the numbers below the bars?

Response

The figure legend has been modified to: “Figure 3. Differentially abundant and unique proteins found in strains O9 and O20. A) Distribution of the proteins found in strain O20 that were found to be unchanged, upregulated, and downregulated compared to the strain O9 of Staphylococcus aureus. The x-axis shows the number of proteins. B) The Venn diagram of proteins identified in the strains O9 and O20.”

Figure 4. Is it correct that DNA binding and RNA binding proteins are shown in different color codes for O9 and O20? If yes, I suggest using the same colors in both graphs.

Response

We modified the figure as suggested by the reviewer, please see below.

Figure 4. What proteins are included in the category “catalytic activity”? Metal- and nucleotide-binding proteins often have a catalytic activity.

Response

Metal and nucleotide-binding proteins are a subdivision of catalytic activity.

Please clarify this relationship in the figure either graphically or in the legend.

Table 2. Did you identify differently downregulated transporters? Does unique mean “only detected in O20” or are these systems truly absent in O9?

Response

We use the term unique, as it is mostly used in proteomics, to report that the protein was only identified in the specific sample (in this case, in the O20). It does not necessarily mean that the protein is not expressed in the other sample, but rather that the protein was not detected by the mass spectrometer in the other sample, which can happen due to the intrinsic limitations of MS-based proteomics, such as low signal of peptides. In our study, we limited this issue by analyzing a high amount/concentration of proteins, and by highly fractionating the sample in a MudPIT strategy, as explained in our methodology.

Author Response

Reviewer 3

I want to thank the authors for the revised manuscript and the answers to my comments. In some cases, I miss an answer or find it incomplete. Thus, I suggest to add information where indicated below.

 Response

We thank the reviewer for encouraging comments and below try to address each point raised

General:

The authors describe a comparison of two Staphylococcus aureus isolates with different sensitivity to ivermectin in vitro. Discussion and interpretation of the results focus on efflux pumps. The topic is of high importance in the light of MRSA infections and of antibiotic resistance in general.

Response

We thank the reviewer for your encouraging comments, and we have responded to each point raised below.

Broad comments:

It would be important to explain the selection of the two isolates used in the study in more detail. How many isolates were tested on ivermectin sensitivity and how do the sensitivities of O9 and O20 compare to the overall distribution? What are the sensitivities of O9 and O20 against classes of antibiotics with known mode of action?

Response

The following text has been added in the materials and methods section of the revised manuscript

Two S. aureus clinical isolates (O9 and O20) were used in the present study. The isolates were isolated and identified as S. aureus from mastitis cases from Lahore, Pakistan. Following initial isolation and identification the isolates were shipped to McGill University, Canada for further analyses. Both strains were sensitive to methicillin and cefoxitin classifying them as methicillin sensitive Staphylococcus aureus (MSSA). The bacteria were grown in Cation-adjusted Mueller Hinton (MH) broth and tryptic soya broth (TSB) (Sigma Aldrich, Canada) at 37°C. There are no standard guidelines developed by the Clinical and Laboratory Standard Institute (CLSI) for the use of IVM against bacteria. Thus, for this study we define the sensitive strain as the one which could be killed by IVM at the concentration of <12 μg/ml versus the resistant strain which was not killed by IVM upto 100 μg/ml concentration.

I was more wondering how the phenotypes of these isolates compare to other mastitis isolates. Please add a statement if IVM resistance is common amongst such isolates or if it is rare. Please add the requested information about resistance/ sensitivity to known classes of antibiotics in addition to methicillin and cefoxitin (especially for agents, which are substrates of the described pumps).

Response

We do not know the seriousness of the problem for IVM resistance in mastitis isolates in general. We have added the resistance profiles to the revised manuscript as follows “Both strains were sensitive to methicillin, and cefoxitin classifying them as methicillin sensitive Staphylococcus aureus (MSSA). In addition, both strains were resistant to tetracyclines, and erythromycin. O9 was sensitive to norfloxacin and ciprofloxacin while O20 was resistant to these two fluoroquinolones.”

The evaluation and discussion of the data concentrates on efflux pumps. It would be highly interesting to know, which other proteins were differently expressed in the two strains (up- and down-regulated). I suggest to add more information on the results and to discuss the findings to speculate about resistance mechanism in a broader sense. A broad analysis might also allow speculation on the mechanism of action of ivermectin. It should be shown that overexpression of the identified efflux pumps is relevant. The authors lists a number of compounds, known to be substrates of the three efflux pumps in the discussion. Please provide susceptibility data to show if sensitivity towards substrates of the pumps are different in the two isolates.

Response

Both up and downregulated proteins in the resistant isolate were considered in the proteomic analysis. We specifically targeted the transporters, since they present an important role in drug resistance, and in this dataset, they were either unique or upregulated. While we agree with the reviewer that in order to comprehend the exact relationship between efflux pumps, IVM and S. aureus, it is a must to block these transporters with known blocking against that would reverse the resistant phenotype. However, due to the pandemic the laboratories are closed, and it is impossible for us to carry out these experiments at this point of time.

Please show how expression levels of the efflux pumps rank in comparison to other up-regulated or unique proteins. I did not ask for blocking efflux pumps but for evidence that pump substrates other than IVM are affected by the overexpression. This relates to my previous comment on antibiotic resistance patterns.

Response

We thank the reviewer for his/her comment and have added the following statement in the discussion section of the revised manuscript.

“Proteins that were only found in one of the samples did not present a fold change. In addition, there were at least 230 proteins with higher fold changes when compared between the two isolates. The group of 230 proteins that were up-regulated had roles in cellular, metabolic processes, biological regulation, efflux pumps for transportation, response to stimulus, localization, cytolysis, cell adhesion and cytokinesis with decreasing order, respectively (supplementary table 1 and 2). However, as efflux pumps are one of the key resistance determinants involved in IVM resistance against different parasites, so they were the focus of this study.”

Specific comments:

p1, lines 30 ff. Significance and Introduction. Please clearly state for which indications ivermectin is approved and what are preclinical activities.

Response

Relevant text added in the introduction.

p1, lines 30. Did ivermectin win the noble price or was it the scientist who discovered ivermectin?

Response

We have now modified the statement as follows “Ivermectin (IVM) is popular as a miracle drug and the scientists who discovered the drug were awarded the Noble Prize in Physiology and Medicine for the year 2015.”

p2, lines 56-59. This part should be moved to the methods section and needs to be justified. Why were these cutoff values applied (see also broad comment 1).

Response

The part has been moved to materials and methods section

p2, lines 67-69. Is this true for S. aureus in general or does the WHO (and others) list MRSA?

Response

The WHO classifies both S. aureus as well as MRSA in the list of bacteria that are of high priority and require new antibiotics, please see link below

https://www.who.int/news/item/27-02-2017-who-publishes-list-of-bacteria-for-which-new-antibiotics-are-urgently-needed

The WHO lists S. aureus as top priority pathogen if it is resistant to methicillin or to vancomycin: “Staphylococcus aureus, methicillin-resistant, vancomycin-intermediate and resistant”. The CDC lists MRSA as a serious threat. Please specify this fact in the manuscript. 

Response

We have modified the text according to the reviewer suggestion as follows “To this end, World Health Organization (WHO) has classified S. aureus resistant to methicillin and vancomycin as one of the microorganisms that immediately require new antibiotics for successful therapy.”

p3, lines 93-94. Should the DMSO content not be kept constant to avoid unspecific effects?

Response

We agree with the reviewer, there was typographical error, so we correct it as follows “IVM dissolved in dimethylsulfoxide (DMSO) (solvent control) was then serially titrated at concentrations of 100, 50, 25, 12.5, 6.25, 3.125 and 1.56 μg/ml.”

This procedure apparently leads to a titration of DMSO. Please specify if the final DMSO content was equal at all concentrations.

Response

We agree with the reviewer that the DMSO concentration is titrated during the IVM titration. In our control (DMSO) group we used the highest amount of DMSO required for to get 100 ug/ml IVM.

Section 2.4. Please add a statement to explain which proteins were isolated (whole proteome or soluble part?).

Response

The following statement was added: “Protein extract of cell pellets (100 mg) from each bacterial strain was digested as previously described [21] and whole proteome analysis was carried out as follows”

Please clarify if the sample preparation results in the isolation of soluble proteins only. It is important to understand if efflux pumps are likely to be extracted or not.

Response

In the sample preparation for the mass spectrometry analysis, a buffer containing 100 mMTris/4% SDS pH 8 is utilized for protein extraction. The addition of a detergent (SDS), and, later, of the solubilization of the peptides in a high concentration Urea (2M) solution improve the solubilization efficiency, enhancing the protein extraction [Faktor, J., Goodlett, D. R., & Dapic, I. (2021). Trends in Sample Preparation for Proteome Analysis. In (Ed.), Mass Spectrometry in Life Sciences and Clinical Laboratory. IntechOpen. https://doi.org/10.5772/intechopen.95962].

We have now modified the text in the manuscript for clarity on this and have added the citation above.

p4, line 142. “used” or similar instead of “inputted”?

Response

Inputted is replaced with “used”.

Figure 1. Please describe the panels in alphabetic order (A, C before B, D).

Response

The panels are now described in the alphabetic order.

p7, lines 217-218. It is not clear what the authors want to say here.

Response

We modified the text of the methods section and the text in the item 3.2 for clarification.

p7, line 218. How does the total number of 2165 identified proteins relate to the total number of proteins in S. aureus?

Response

The following sentence has been added: “A total of 2,165 proteins were identified in the dataset of strains O9 and O20, representing about 75% of the proteins from the reference proteome in UNIPROT database for S. aureus strain NCTC/PS47 (proteome ID: UP000008816).”

Figure 3. What is the unit of the numbers below the bars?

Response

The figure legend has been modified to: “Figure 3. Differentially abundant and unique proteins found in strains O9 and O20. A) Distribution of the proteins found in strain O20 that were found to be unchanged, upregulated, and downregulated compared to the strain O9 of Staphylococcus aureus. The x-axis shows the number of proteins. B) The Venn diagram of proteins identified in the strains O9 and O20.”

Figure 4. Is it correct that DNA binding and RNA binding proteins are shown in different color codes for O9 and O20? If yes, I suggest using the same colors in both graphs.

Response

We modified the figure as suggested by the reviewer, please see below.

Figure 4. What proteins are included in the category “catalytic activity”? Metal- and nucleotide-binding proteins often have a catalytic activity.

Response

Metal and nucleotide-binding proteins are a subdivision of catalytic activity.

Please clarify this relationship in the figure either graphically or in the legend.

Response

We have modified the figure legend accordingly as follows “Figure 4. Pie charts of the functional annotation of all proteins identified in strain O20 (A) and strain O9 (B). GO terms assigned are associated with molecular function from UNIPROT-GOA. Metal and nucleotide-binding proteins are a subdivision of catalytic activity.”

Table 2. Did you identify differently downregulated transporters? Does unique mean “only detected in O20” or are these systems truly absent in O9?

Response

We use the term unique, as it is mostly used in proteomics, to report that the protein was only identified in the specific sample (in this case, in the O20). It does not necessarily mean that the protein is not expressed in the other sample, but rather that the protein was not detected by the mass spectrometer in the other sample, which can happen due to the intrinsic limitations of MS-based proteomics, such as low signal of peptides. In our study, we limited this issue by analyzing a high amount/concentration of proteins, and by highly fractionating the sample in a MudPIT strategy, as explained in our methodology.